# Effect of Particles Size on Dielectric Properties of Nano-ZnO/LDPE Composites

**DOI:** 10.3390/ma12010005

**Published:** 2018-12-20

**Authors:** YuJia Cheng, Liyang Bai, Guang Yu, Xiaohong Zhang

**Affiliations:** 1Mechanical and Electrical Engineering Institute, University of Electronic Science and Technology of China, Zhongshan Institute, Zhongshan 528402, China; chengyujia1068@163.com; 2State Grid Tianjin Wuqing Electric Power Supply Company, Tianjin 301700, China; mdpinao@126.com; 3Key Laboratory of Engineering Dielectrics and Its Application, Ministry of Education, Harbin University of Science and Technology, Harbin 150080, China; x_hzhang2002@sina.com

**Keywords:** LDPE, ZnO, particles size, dielectric properties

## Abstract

The melt blending was used to prepare 3 wt% ZnO/low density polyethylene (ZnO/LDPE) nanocomposites in this article. The effect of different inorganic ZnO particles doping on the dielectrical property and crystal habit of LDPE matrix was explored. The nanoparticles size was 9 nm, 30 nm, 100 nm, and 200 nm respectively. Scanning electron microscope (SEM) was used to characterize ZnO nanoparticles whereas differential scanning calorimetry (DSC) was used to make thermal characterization of the samples. Besides, the AC (alternating current), DC (direct current breakdown characteristics and electrical conductivity of the nanocomposites was studied in this article. The experimental results showed that nano-ZnO/LDPE composites had the advantages such as small crystal size, high crystallization rate and crystallinity owing to nano-ZnO particles doping, when doping nano-ZnO particles size was 30 nm, the ZnO/LDPE nanocomposite crystallinity crest value 39.77% appeared. At the mean time, the DC and AC breakdown field strength values of composites were 138.0 kV/mm and 340.4 kV/mm respectively. They were the maximal values which improved 8.24% and 13.85% than LDPE. The AC breakdown field strength of samples decreased with specimen thickness increase. The DC breakdown field strength of LDPE and ZnO/LDPE composites were greater than AC breakdown field strength. From the conductivity experimental result it could be seen that when the experimental temperature and electric field intensity rose, the current density and conductivity of ZnO/LDPE composites increased with the enlargement of ZnO particles size. But the values were less than which of LDPE.

## 1. Introduction

With the rapid development of economy and industrialization in China, the electrical source increases. The electrical insulation regular work is the guarantee of electric system reliable operation. How to protect the high efficiency and high quality energy transmission becomes the first imperative problem to resolve. Low density polyethylene (LDPE) is one kind of nonpolar macromolecule polymeric material which has some advantages such as high insulation resistance, low permittivity, and low loss and low impact of temperature. Therefore, LDPE is widely used as base material of electric power cable insulation system. The cable would be affected by multiple parameters such as electricity, heat and machinery in long-time running process which would cause the cable insulating property degradation even insulation failure. For this reason, how to improve the cable insulation electric property is concerned by many research scholars [1,2,3]. Owing to the nanoparticles have some excellent chemical and physical properties such as small specific area and high apparent activation energy, which provides a new path of cable insulation strength improvement to researchers [4,5,6,7,8]. Nano-ZnO was one kind of new inorganic product. It was an elaborate compound semiconductor material with high performance and large energy gap, and it had nonlinear characteristics [9,10,11,12]. Compared with common ZnO, nano-ZnO had many advantages such as surface effect, volumetric effect, quantum effect, and dielectric confinement effect. Owing to the excellent properties, the nano-ZnO was widely used. According to the blending technology, the inorganic semiconductor material with nonlinear characteristic could be added into the polymeric matrix. The nonlinear composite insulating material of which conductivity and dielectric constant changed with electric field intensity could be prepared. This nonlinear insulation material had the function of grading the electric field in complex insulation structure, which could improve the dielectric strength of insulation system. Therefore it had a broad space such as being used in insulation system of extreme high voltage DC (direct current) bushing and ultrahigh voltage direct current cable. At the same time, the different nanoparticles size would make large influence to polymer composites performance. As the nanoparticles size decreased, the specific area increases and the apparent activation energy rose, which would cause the agglomeration between nanoparticles easily [13,14,15]. When the nanoparticles dispersed uniformly in polymer and the sample volume was the same as nano-doping mass fraction, the different particles size would cause the difference of particles distance and particles with polymer crystalline distance. Therefore different particles size affected the micromechanism and macroscopic properties of polymer nanocomposite [16,17]. Moreover, the effect of different doping nano-ZnO particles size on polymeric crystal habit and dielectrical property could be further explored, which was based on the present research production of nano-ZnO/LDPE insulation. It could provide the theory evidence in high dielectrical property insulation material preparation. 

This article relied on National Natural Science Foundation of China which was “Investigation on meso-morphology and mechanism inhibiting space charge in polymer/inorganic micro-nano composites based on the coordination effect of micro- and nano-filler”. The nano-ZnO with different particles size was doped into polyethylene matrix, of which ZnO doping mass fraction was 3 wt%. The ZnO/LDPE nanocomposite was prepared by melt blending. From microscopic test and microcosmic test, the disperse state of different size ZnO particles doping into the matrix and polymeric composites crystallization could be explored. The composite AC (alternating current)/DC (direct current) breakdown characteristics were discussed in this article. Besides, the variation characteristics of current density and conductivity under different electric field intensity and temperatures was studied, which could explore the impact mechanism of different inorganic nanoparticles size on internal electrical charges transport property of nano-ZnO/LDPE composite.

## 2. Materials and Methods

### 2.1. Sample Preparation

The main experimental materials and equipments were LDPE (Da Qing Petrochemical, Da Qing, China) of which density portion was 0.910~0.925 mg/cm^3^; nano-ZnO (Beijing Decaux Science and Technology Ltd., Beijing, China) of which particles size was 9 nm, 30 nm, 100 nm, and 200 nm; 1010 antioxygen; rheometer; and platen press.

The melt blending was used to prepare different particles size nano-ZnO modified polyethylene composite in this experiment. The same mass fraction 3 wt% but different particles size nano-ZnO was doped into LDPE, which was dealt with rheometer melt blending in 20 min. The temperature was set to 140 °C. The ZnO/LDPE sample, of which thickness was 65 ± 10 μm, 130 ± 10 μm, 195 ± 10 μm and 200 μm, were presses by platen press. The detailed preparation was shown in Figure 1.

### 2.2. Morphology Characterization of Different Particles Size ZnO

In order to discuss the influence on different size nano-ZnO particles doping to LDPE performance, this nano-ZnO particles were made the characterization by scanning electron microscope (SEM, Hitachi S-4800, Tokyo, Japan), which could observe the doping particles size. Firstly, a spot of different particles size ZnO powder was smeared on conducting resin respectively. Then, the samples are observed by SEM. The observations were shown in Figure 2.

From Figure 2 it can be seen that the inorganic nano-ZnO particles size were 9 nm, 30 nm, 100 nm, and 200 nm respectively. When the nano-ZnO particles size was fairly small, the size distribution range were very little, accordingly, all the particles size was basically identical. While the particles size was superior, the size distribution range would increase thereupon. When the particles size was less than 200 nm, all the particles shape was round-likely roundness. When the particles size was 200 nm, the particles were divided into two parts, one was round-likely roundness, the other was rodlike. The shape of ZnO particles was irregular at present.

### 2.3. Crystal Analysis of ZnO/LDPE Composites

In order to explore the crystallization and crystallinity of ZnO/LDPE composite, the differential scanning calorimeter (DSC, Mettler Toledo, Zurich, Switzerland) was used to analyze the composite crystallization and test the relevant crystal parameter in this experiment. Firstly, the samples were placed in nitrogen condition of which flow was 150 mL/min. The rate of heating and cooling was 10 °C/min in this experiment. In order to avoid the deviation caused by samples heat history to experimental result, the samples were warmed up to 140 °C which made the material full melt. Secondly, the rate declined to 25 °C.

The enthalpy change was recorder, from which the crystal temperature *T*_c_ and the width of exothermic crystalline peak Δ*T*_c_ could be obtained. Then the composite was warmed up to 140 °C again. The composite enthalpy change during continuous heating was recorded, from which could obtain the DSC temperature *T*_m_ and the heat of fusion Δ*H*_m_ automatically [18]. The melting temperature *T*_m_ was determined by melting end temperature. The higher the melting temperature, the bigger the spherulite size was [19]. The temperature, of which crystalline nucleation rate was highest, was the crystalline peak temperature *T*_c_. The width of exothermic crystalline peak Δ*T*_c_ was based on the width of half peak in exothermic peak. The DSC test charts of different composite and LDPE were shown in Figure 3.

From these parameters and Formula (1), the crystallinity of LDPE and all ZnO/LDPE composites could be calculated.
(1)$$X_{c}=\frac{\Delta H_{m}}{\left(1-\omega \right)H_{0}}\times 100\%$$

In Formula (1), Δ*H*_m_ was the material melting heat (J/g). *H*_0_ was the melting heat of material fully crystallization, which of LDPE was 293.6 J/g. *ω* was the mass fraction of different particles size ZnO doping [20].

The isothermal crystallization and melting procedural parameters of composite detected by DSC experiment were shown in Table 1.

From Table 1, it could be seen that the LDPE crystallinity and melting heat would change as the ZnO particles size change. After different particles size ZnO doping, the melting temperature *T*_m_ of nano-ZnO/LDPE decreased, which could indicate the small crystal size of composites. The Δ*T*_c_ of composites decreased. Crystalline nucleation rate could be characterized by the width of Δ*T*_c_. Comparing with pure LDPE samples, the crystalline nucleation rate of ZnO/LDPE composites increased. The crystallinity of ZnO/LDPE composites with different particles size would increase except the composite with 200 nm particles size of which extent was different. The crystallinity specific relationship was 30 nm > 9 nm > 100 nm > LDPE > 200 nm. When doping nano-ZnO particles size was 30 nm, the ZnO/LDPE nanocomposite crystallinity crest value 39.77% appeared.

The cause might be the filler particles size was small, which could be used as nucleating center by being doped into the polymeric matrix as nucleating center. Some interface regions were taking shape in composite. Meanwhile, the ZnO particles acted as heterogeneous nucleation. LDPE molecules made the crystallization around the additive particles which could improve the crystallinity of ZnO/LDPE composites [21,22]. In the wake of doping ZnO particles size increased, the hererogeneous nucleation faded. When the mass fraction of doping ZnO was identical, the smaller the particles size was, the more the ZnO particles in unit volume sample would be. The heterogeneous nucleation centralities increased, which improved the composites crystallinity. But the composites crystallinity with 9 nm particles size was less than which of 30 nm. The cause might be 9 nm ZnO particles size was very little and the polarity was strong, which produced the agglomeration effect. Therefore the crystallinity was lower than which of 30 nm. When the particles size increased to 200 nm, the nanometer characteristics of inorganic ZnO nanoparticles weakened, which caused the crystallinity decrease.

## 3. Results and Discussion

### 3.1. Atrernating Current Breakdown Test and Analysis

DC (direct current) and AC (altrernating current) ower frequency breakdown system was used in this experiment, which boosted in uniform speed with 1 kV/s until the material was breakdown. The breakdown potential value U could be fetched and the breakdown point thickness could be measured. Being combined with the formula *E* = *U/d*, the samples breakdown strength could be calculated. The thickness of test specimen was about 65 ± 10 μm, 130 ± 10 μm, 195 ± 10 μm. The breakdown field strength of each sample should be test for 20 points. In order to eliminate the material history effect, the samples must make the pretreatment before this experiment. The vacuum drying oven temperature was set to 80 °C, in which the samples were placed for 24 h. In case the samples creepage discharge happened, the whole electrode system and samples must be dipped in cable compound together. The voltage transformer boosted in uniform speed until the short circuit gave an alarm. At present, the impressed voltage was the breakdown voltage of samples, of which data was recorded. The breakdown point thickness d was measured. After that, the breakdown field strength of different particles size ZnO/LDPE composites could be obtained. The MINITAB statistical software was used to make the Weibull distribution curve. The parameters in Weibull distribution included form parameter *β* and scale parameter *E*. The parameter *β* represented data spread degree and *E* represented Weibull breakdown field strength. The symbol N represented the testing times of each sample. The experimental results were shown in Figure 4, Figure 5 and Figure 6.

From the experiment results of Figure 4, it can be seen that the breakdown field strength of ZnO/LDPE composites firstly increased and then decreased with ZnO particles size accretion. When the nano-ZnO particles’ sizes were 9 nm, 30 nm, and 100 nm, the composites breakdown field strength were higher than which of pure LDPE. The highest value 138.0 kV/mm appeared when the particle size was 30 nm. It was 8.24% higher than pure LDPE. The breakdown field strength of ZnO/LDPE composites was lower than which of LDPE when the particle size was 200 nm. It was 3.8% lower than pure LDPE.

When the samples thickness was 130 ± 10 μm, the breakdown field strength Weibull distribution of ZnO/LDPE and LDPE with different particles size were shown in Figure 5. From Figure 5, when the samples thickness was 130 ± 10 μm, the breakdown field strength variation trend of ZnO/LDPE composites with different ZnO particles size was the same as the 60 ± 10 μm thickness samples. While the ZnO particles size was 9 nm and 30 nm, the breakdown field strength of composites was higher than which of LDPE. While the ZnO particles size was 100 nm and 200 nm, the breakdown field strength of ZnO/LDPE composites was less than which of LDPE. The highest value 122.7 kV/mm appeared when the ZnO particle size was 30 nm, which was 8.3% higher than pure LDPE.

When the samples thickness was 195 ± 10 μm, the breakdown field strength Weibull distribution of ZnO/LDPE and LDPE with different particles size was shown in Figure 6. From the experimental result, when the samples thickness was 195 ± 10 μm, the breakdown field strength of ZnO/LDPE composites, of which particles size was 9 nm, 100 nm and 200 nm, was less than LDPE. While the ZnO particles size was 30 nm, the breakdown field strength of composites was higher than LDPE, which improved 1.7%.

The reason might be the nanoparticles doping made the contribution of heterogeneous nucleation. The crystallinity of nano-ZnO/LDPE composites raised and the interfacial structure were complicated [23,24,25,26]. There were some traps in interface, which could catch the carrier effectively and reduce the carrier mobility in the samples. After carrier trapping, it was hard to make detrapping. The interface localized state was formed within the samples. On the one hand the charge trapping diminished the carrier drift velocity, on the other hand the Coulomb field, which was produced by like charges, offset partial external electric field. It could reduce the local electric field intensity within samples. The charge mobility decreased with the quantity of electric charge injected by electrode decrease. By the effects of these factors, the carrier quantity and mobility of nanocomposite fell drastically. Owing to the difference of interface region thickness, the average distance between particles and ZnO particle quantity in unit volume formed by different particles size ZnO/LDPE composite, the breakdown field strength of composites was different. When the nano-ZnO particles size was smaller, the composite trap density was high and the trapping level was deep. The carrier would be catch by the traps in the process of transfer, the interstitial volume diminished and the free travel elongated, by which the carrier could not pick up the kinetic energy easily. Therefore, the breakdown field strength of composites was higher. When the nano-ZnO particles size was 9 nm, the composites breakdown field strength was lower than which of 30 nm. The cause might be the 9 nm ZnO particles quantity within the samples was large in same mass fraction. Moreover, the particles polarity was very strong and the agglomeration effect was obvious at this time. Therefore the breakdown field strength of composites was the highest with the 30 nm ZnO particles doping [27]. When the samples thickness was 65 ± 10 μm, the breakdown field strength of 9 nm and 30 nm particles size composites was higher than which of LDPE obviously. It could confirm the definition of 1~100 nm nanoparticles size. When the particles size is larger than 100 nm, some advantages of nanoparticles doping, such as small size effect and surface effect of nano-additive completely are not obviously. Therefore the samples breakdown field strength value would decrease when the nano-ZnO particles size was 200 nm.

Figure 7 showed the AC breakdown field strength of various particles size ZnO/LDPE composites in different samples thickness.

From Figure 7 it can be seen that each composite and LDPE threshold field strength increased with samples thickness decreases. The reduction extent of breakdown field strength was different in various particles size composite. Moreover, the thicker the samples’ thickness was, the more the breakdown field strength decreased. LDPE, for example, the breakdown field strength reduced by 11% with the thickness increased from 65 μm to 130 μm. The breakdown field strength reduced by 25% with the thickness increased from 130 μm to 195 μm.

The cause might be the breakdown of ZnO/LDPE composites was not a purely electric breakdown, which was composed by electric breakdown and thermal breakdown after power frequency alternating voltage imposed to composites. When the samples’ thickness was thinner, the samples’ breakdown was electric breakdown mainly. Therefore the breakdown strength of various particles size composites was related with interfacial area densities and volume. The electric field local heterogeneity affected much of electric breakdown. When the samples’ thickness increased, the heat quantity which generated by composites loss under electric field could not dissipate effectively [28]. The heat accumulation was serious, which caused the insulating ability loss in dielectric. The composites breakdown strength would be greatly affected by thermal breakdown. At the mean time, the temperature and voltage actuation duration were main factors affecting samples thermal runaway. Besides, the thicker the samples thickness was, the harder the heat radiation would be. The effect of thermal breakdown would be obvious which made the breakdown field strength decrease more. On the other hand, the composites breakdown was one kind of weak point breakdown. When the thickness increased, the samples volume grew larger which cause the weak points increase, this factor causes the breakdown field strength of composites decrease.

### 3.2. Direct Current Breakdown Test and Analysis

The DC (direct current) electric field was used to make the breakdown characteristics test in ZnO/LDPE composites. The samples thickness was 65 ± 10 μm. The other experiment conditions were identical to AC breakdown. With the analysis of MINITAB statistical software, the Weibull distribution curve could be made in Figure 8.

From Figure 8 it can be seen that the DC breakdown field strength of LDPE and different particles size ZnO/LDPE composites was higher than which of AC breakdown field strength. The change regulation with particles size was similar to AC breakdown. The specific comparison was 30 nm > 9 nm > 100 nm > LDPE > 200 nm. The DC breakdown field strength of 30 nm ZnO particles size composites was the highest, which was 13.8% higher than LDPE. The cause might be the AC breakdown was conducted in alternating electric field. The carriers moved back and forth between two electrodes. These electrons would accelerate under the action of external electric field. Moreover, they could make the combination with crystal lattice, which caused lattice vibration. The energy of electric field was transferred to crystal lattice. With the external electric field increasing, the energy of electron taking from electric field would be greater than the loss one of electron transferring to crystal lattice. The kinetic energy of electron would rise higher and higher. When the energy of the electrons exceeded a certain value, the combination of electrons with lattice vibration would make the ionization and new electrons appearing. The free electrons’ quantity increased rapidly, which caused the samples breakdown. Under the condition of DC electric field, the carriers move to positive electrode. The impact ionization seldom appeared throughout the process. The DC breakdown field strength was higher than which of AC. The DC breakdown field strength change regulation of different particles size composites was identical to which of AC. Tanaka suggested that the carrier transport was carrier density, trap density and depth determined under application of external electric field. The breakdown strength was related to carrier migration rate and transport path. The smaller size nanoparticles introduced the traps, which restricted the carrier mobility and improved the breakdown field strength. When the particles size increased, the interface effect would weaken, which declined the breakdown field strength of composites.

### 3.3. Conductivity Test and Analysis

The DC (direct current) high voltage source was selected in this experiment. The composite current was measured by picoammeters. By analyzing the data, the relationship curves between electric field intensity with conductivity and current density could be obtained. The deferent crest value of testing DC high voltage was 15 kV and the supreme testing accuracy of picoammeters was 10^−15^ A. The internal current stable value of samples could be measured. The relationship curves between current density with impressed voltage of different particles size ZnO/LDPE composites and LDPE were shown in Figure 9a,b.

The conductivity measurement result of ZnO/LDPE composites with different particles’ size were shown in Figure 9. Figure 9a was the relationship curve between electric field intensity with current density. Figure 9b was the relationship curve between conductivity with electric field intensity which was drawn by log-log scale.

From current density curve in Figure 9a, it can be seen that the LDPE current density increased with impressed voltage under the test voltage. But this curve did not display linear trend, which showed three different gradient regions in range of test voltage. This phenomenon was similar with which was described by space charge limited current theory of solid dielectric [29]. When the external field strength was low, the dielectric current accorded with Ohm’s law, which is called Ohm current range. The electrons injected by electrode increased with the external electric field rose, which caused the space charge accumulation within dielectric. Moreover, the space charge limited current arose. The trapping of dielectric traps to electrons would reduce the carrier mobility, which made the considerable increase in electronic current not happen. This area was traps active region, which disobeyed Ohm’s law. When the external field strength continues to increase, the dielectric space charge was growing fast. When the traps in samples were filled up by electrons, the power surge would arise within samples. The current density of ZnO/LDPE composites increased in exponential form with electric field intensity but which were all different. The current density of different particles size ZnO/LDPE composites increased with ZnO particles size in same field strength. But the current density of each composite was less than LDPE. It can be seen from Figure 9b that all composites and LDPE conductivity increased with the particles size under same electric field intensity. But all of composites were less than LDPE. The conductivity curve of each sample could be divided into two parts roughly. When the field strength was lower, the conductivity did not increase markedly as electric field increase. When the field strength increased to threshold field strength, the conductivity increased significantly with field strength improve. All composites threshold field strength was different. Except the 9 nm particle size ZnO composite, the other composites threshold field strength decreased gradually with the particles size increase.

The cause of this phenomenon could be summarized as follows. The steady conduction current was accomplished by samples inner carrier movement in dielectric. The current carrier is composed of electron and hydronium for LDPE. The electron was injected to the samples by electrode in condition of voltage impressed. When the electric field intensity increased, on the one hand the electron and hydronium in dielectric would sustain a large electrical stimulation, on the other the injected electron increased from electrode to dielectric, which caused the carrier concentration and mobility increase. The dielectric conductivity was decided by carrier concentration and mobility [30,31]. Under application of an electric field, the dielectric carrier freemove would create a current, which could be expressed to Formula (2).
(2)γ=nqμ

In Formula (2), *n* expressed carrier number. *q* expressed the quantity of electric charge and *μ* expressed carrier mobility.

The conductivity and current density of composites increased according to Formula (2). After ZnO doping into LDPE, the ZnO heterogeneous nucleation would form the interphase between additives with LDPE. There were a large number of traps in interface. It could capture the carrier which reduced the carrier concentration and mobility. The conductivity and current density of composites would decrease. Besides, the smaller the nano-ZnO particles size was, the relative volume of interphase structure formed by nanoparticles and matrix in whole composites would increase. The deep trapping level would strengthen the carrier trapping. Accordingly the conductivity and current density of composites increased with the enlargement of ZnO particles size. The relationship curve of conductivity and electric field intensity was divided into two parts. When the electric field intensity was lower, the trapping of composites traps would make the conductivity increase indistinctly. When the field strength increased to a certain level, the traps would be filled up. It would weaken the restriction to carrier and the conductivity increased. The depth and density of traps depended on the size of threshold field strength. With the deep trapping level and high density, the traps were hard to be filled up. The threshold field strength would increase accordingly. All composite threshold field strength would decrease with nano-ZnO particles size increase in samples. But 9 nm ZnO/LDPE composite showed the reversal. That might be the very small size of particles caused obvious agglomeration, which reduced the exceptional property of nanoparticles.

In order to discuss the effect of different experimental temperature on composite electrical conductivity characteristics, in this article, the 20 kV/mm field strength was used to test the current density and conductivity of composite under 25 °C, 45 °C, 60 °C, 75 °C, and 90 °C respectively. The test results of composite electrical conductivity characteristics were shown in Figure 10a,b.

Figure 10a showed the relationship curve between current density with temperature of LDPE and composites. From the experimental result of Figure 10a, the composites current density increased with temperature raises. When the experimental temperature was lower, the increasing range of current density was small. The temperature had little influence on current density. When the experimental temperature was higher, the temperature had more important effect on conductivity. Under the same temperature, the composite current density still increased with the enlargement of ZnO particles’ size. But the value was less than which of LDPE.

Figure 10b showed the relationship curve between conductivity with temperature. From the experimental result of Figure 10b, the composites’ conductivity increased with temperature raises. The increasing ranges were identical. But the composites’ conductivity value was less than which of LDPE. It indicated that the heat shock jump process consisted in composite electronic conduction mechanism. The analysis might be the temperature increment caused the dielectric carrier energy increment. The carrier could overcome the barriers between energy bands easily in process of transfer, which increased the carrier concentration, current density, and conductivity. With the same field strength, the temperature had much effect on the dielectric ionic conductance which could be divided into intrinsic ionic conductance and weakly bound impurity ionic conductance. In a condition of low temperature, the weakly bound impurity ionic conductance existed in dielectric mainly. In condition of high temperature, the intrinsic ionic conductance started to appear in dielectric owing to the polymeric disassociation. In high temperature, the further increase of carrier concentration would cause the phenomenon, in which the increasing range of current density would be greater with temperature increase [32,33].

## 4. Conclusions

(1)From SEM experimental result it can be confirmed that the doping particles size of ZnO was 9 nm, 30 nm, 100 nm, and 200 nm respectively. From DSC experimental result, in the wake of particle size-increase, the nanoparticles’ dispersity in the matrix decreased slightly. The crystallinity of ZnO/LDPE composites with different particles’ size would increase except the composite with 200 nm particles’ size. When the mass fraction of ZnO particles doping was 3 wt%, the 30 nm composite crystallinity was the highest of which degree was 39.77%.(2)From the breakdown properties result, the DC and AC breakdown field strength of LDPE and various composites firstly increased and then decreased with the increase of ZnO particles size. The breakdown field strength of ZnO/LDPE composites decreased with the specimen thickness increase. The samples DC breakdown field strength was higher than AC breakdown field strength. When the ZnO particles’ size was 30 nm, the AC and DC breakdown field strength of ZnO/LDPE composites were highest, with values of 138 kV/mm and 340.4 kV/mm respectively. They increase 8.24% and 13.85% respectively compared with which of LDPE. All of them were lower than LDPE.(3)From the experimental result of different composites electrical conductivity characteristics it can be confirmed that the current density and conductivity of ZnO/LDPE composites would decrease with the diminution of ZnO particles size under different electric field. All of them were lower than LDPE. Under the condition of different temperatures, the current density and conductivity of LDPE and various composites increased linearly with temperature rise. The current density and conductivity of composites increased with the enlargement of ZnO particles size in the same temperature. But all of them were less than which of LDPE.

## Figures and Tables

**Figure 1 materials-12-00005-f001:**
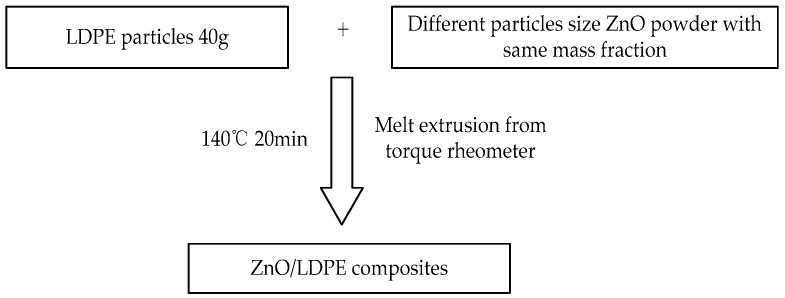
Preparation of 3 wt% ZnO/ low density polyethylene (LDPE) composite.

**Figure 2 materials-12-00005-f002:**
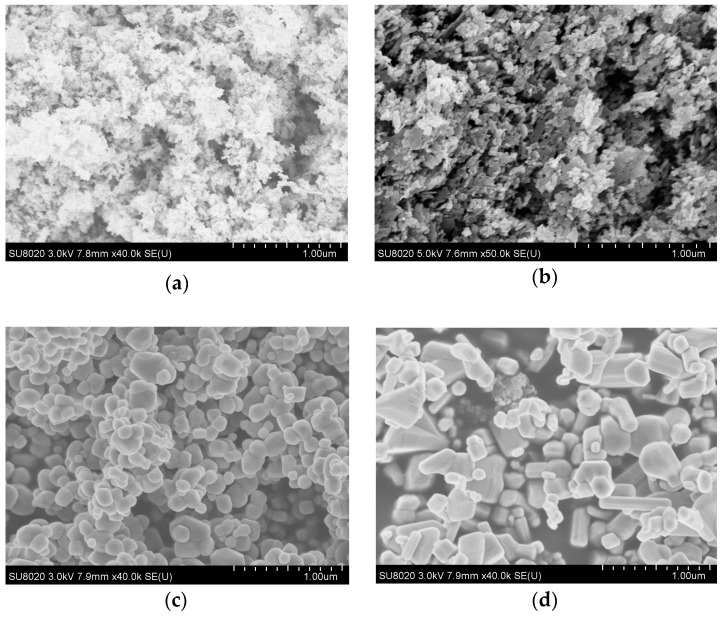
Morphology characterization of ZnO with different particle sizes. (**a**): SEM image of 9 nm ZnO particles; (**b**): SEM image of 30 nm ZnO particles; (**c**): SEM image of 100 nm ZnO particles; (**d**): SEM image of 200 nm ZnO particles.

**Figure 3 materials-12-00005-f003:**
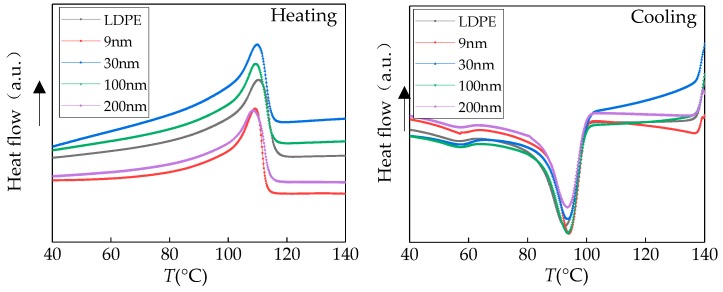
DSC traces of the composite materials.

**Figure 4 materials-12-00005-f004:**
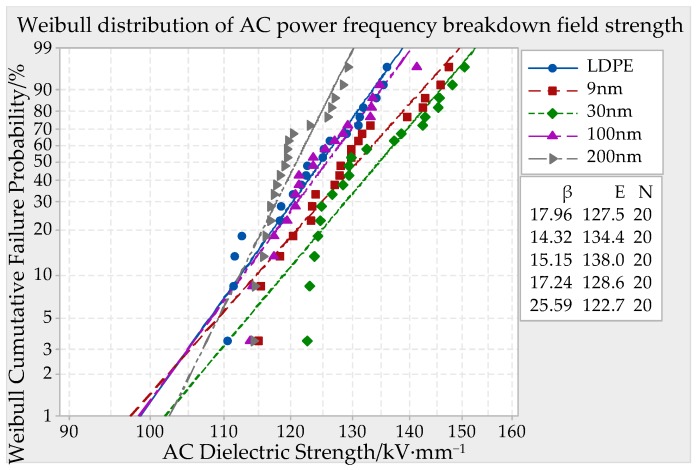
AC (altrernating current) Breakdown field strength of different particle size ZnO/LDPE composite material with a thickness of 65 ± 10 μm.

**Figure 5 materials-12-00005-f005:**
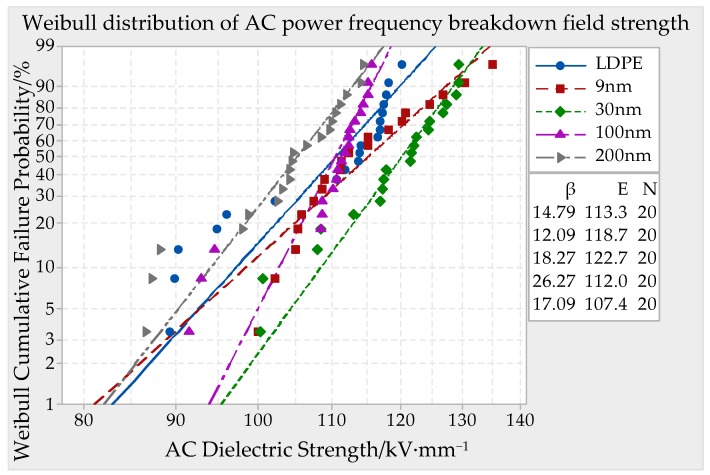
AC (altrernating current) Breakdown field strength of different particle size ZnO/LDPE composite material with a thickness of 130 ± 10 μm.

**Figure 6 materials-12-00005-f006:**
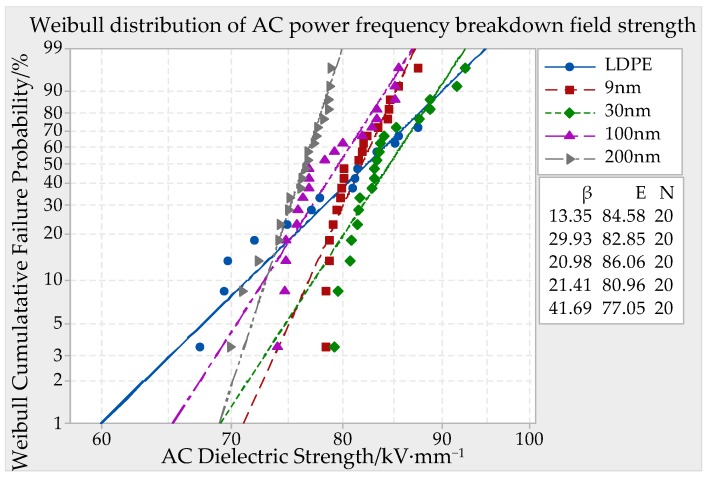
AC (altrernating current) Breakdown field strength of different particle size ZnO/LDPE composite material with a thickness of 195 ± 10 μm.

**Figure 7 materials-12-00005-f007:**
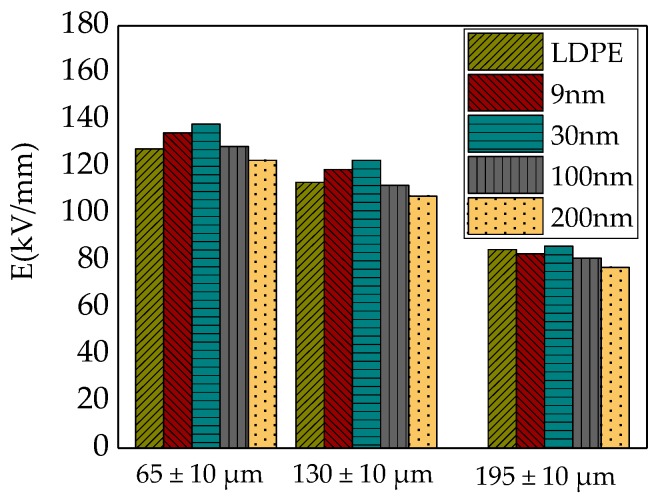
Comparison of the strength of AC (altrernating current) breakdown field of different specimen thickness.

**Figure 8 materials-12-00005-f008:**
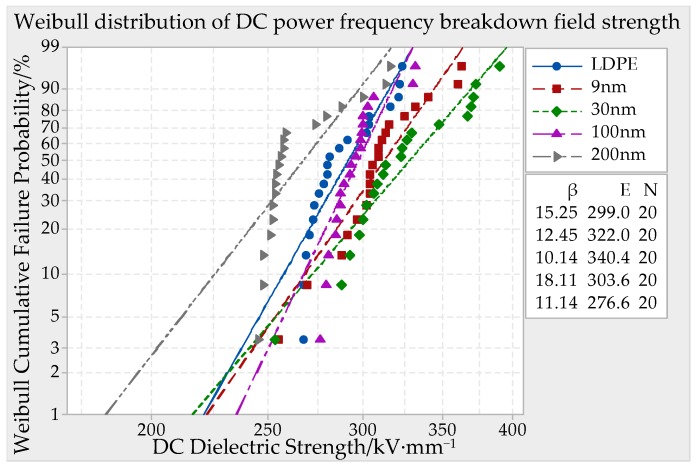
DC (direct current) breakdown field strength of ZnO/LDPE composites with 65 ± 10 μm thickness.

**Figure 9 materials-12-00005-f009:**
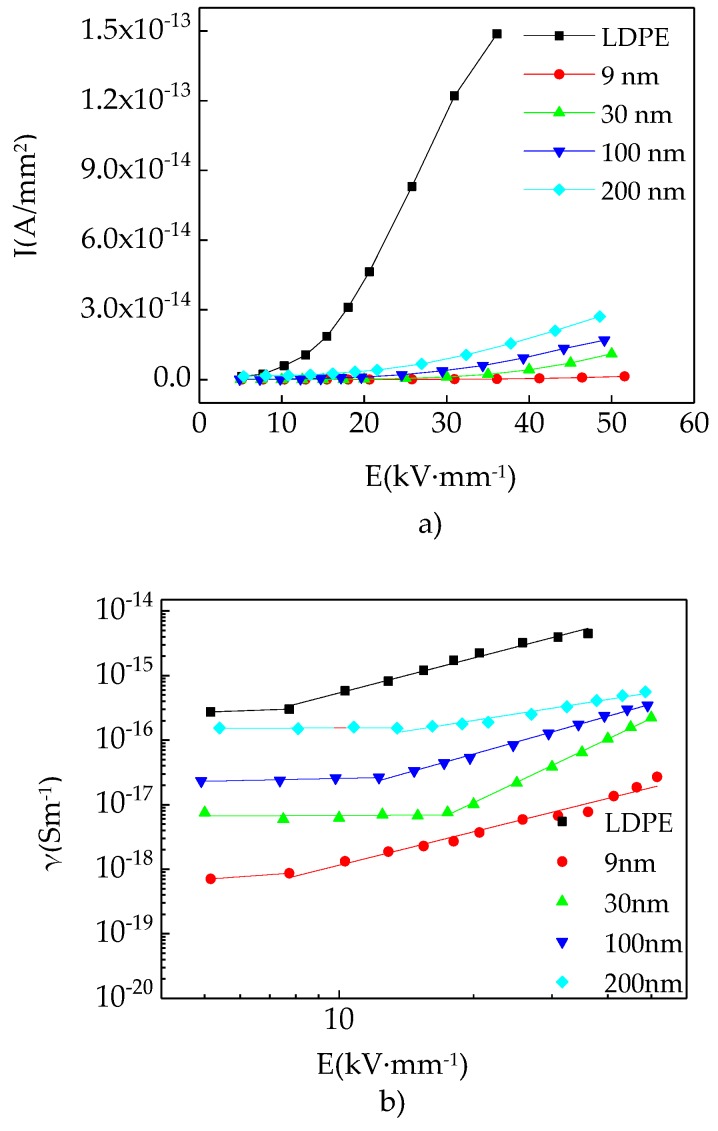
Conductivity curves of different particle size ZnO/LDPE composites. (**a**): The relationship between conductivity current and field intensity for samples (**b**): The relationship between conductivity and field intensity for samples.

**Figure 10 materials-12-00005-f010:**
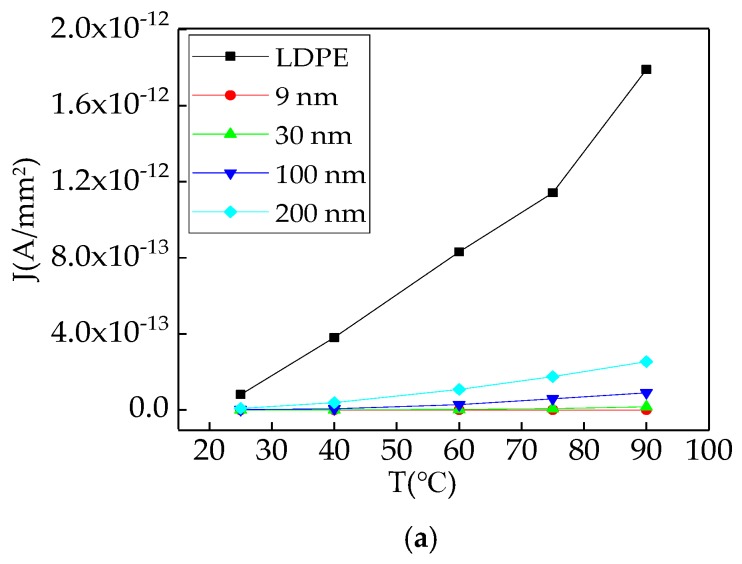
Electrical conductivity characteristics of different particle size ZnO/LDPE composites with temperature variation curve. (**a**): The relationship between conductivity current and temperature for samples. (**b**): The relationship between conductivity and temperature for samples.

**Table 1 materials-12-00005-t001:** Isothermal crystallization and melting process parameters.

Sample	Crystalline Peak Temperature *T*_c_/°C	Melting Temperature *T*_m_/°C	Width of Exothermic Crystalline Peak Δ*T*_c_/°C	Crystallinity *X*_c_/%
LDPE	93.24	109.78	8.54	34.26
9 nm	94.37	108.92	7.02	37.49
30 nm	94.12	109.24	6.22	39.77
100 nm	94.25	108.73	7.69	37.43
200 nm	93.90	108.18	6.90	32.21

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
