# Peer review of "Effect of Particles Size on Dielectric Properties of Nano-ZnO/LDPE Composites"

_materials, 2018, doi:10.3390/ma12010005_

Round 1

Reviewer 1 Report

Comments

The aim of the present work should be cleared out. Why ZnO? References needed.

The filler concentration must be numerically defined. e.g. wt%?

The DSC curves should be shown in heating and cooling cycles in order to understand influence of nanoparticles to crystallization and melting processes.

The technology of electric contacts should be described. Contacts for ionic or electronic conductivity?

All discusion should be revised. The conductivity and breakdown mechanisms is not explained correctly: e.g. if term"traps"is used then it is necessary explain what kind of traps. Is it impuritity traps in semiconductor ZnO?

In composite of polymer and ZnO particles, the percolation currents in polymer or ZnO should be discussed separetely. For ionic and electronic conductivities.

English should be corrected from title to end. 

Scientific definitions should be corrected. What do You mean "particle polarity", electric field intensity", "interphase relative volume"?

First sentence in abstract:"This article uses melt blending..." ???

Author Response

Author Response:

First of all, I’d like to say thanks to the expert’s comments on the manuscript. I feel honored to gain the expert’s encouragement. I agree with the expert on the comments and revision suggestion. I have modified the manuscript and improved the language of the manuscript, the detailed modifications are shown as follows:

Detailed Modifications:

1.        Nano-ZnO is one kind of new inorganic product. It is an elaborate compound semiconductor material with high performance and large energy gap, and it has  nonlinear characteristics. Compared with common ZnO, It has many advantages such as surface effect, volumetric effect, quantum effect and dielectric confinement effect. Thanks to the excellent properties of nano-ZnO, it is used widely. According to the blending technology, the inorganic semiconductor material with nonlinear characteristic could be added into the polymeric matrix. The nolinear composite insulating material of which conductivity and dielectric constant changed with electric field intensity could be prepared. This nonlinear insulation material has the function of homogenizing the electric field in complex insulation structure, which could improve the dielectric strength of insulation system. Therefore it has a broad space such as being used in insulation system of extreme high voltage DC bushing and ultrahigh voltage direct current cable. And nano-ZnO is selected in this paper.

2.       The amount of ZnO doping is 3wt% in this article, which is marked in abstract, samples preparation and comment section of figure 1.

3.       In this article, the DSC experiment is implemented based on the advices of reviewers. The heating and cooling DSC curve of composites is added into this article.

4.       Under high electric field, the solid dielectric loss the electric insulating ability, which made the isolated mode mutated into conductive mode, it was breakdown in solid dielectrics. In uniform electric field, the ratio of breakdown potential and dielectric thickness is called the breakdown field strength. The electrode directly contact with sample surface in breakdown experiment. Under high electric field, the electronic conduction appears obviously within the dielectric owing to the electronic field-emission in electrode.

5.       The further explanation of “traps” in this article: the nanoparticles doping has an influence on polymer trap characteristics. Firstly, the nanoparticles doping play a role in heterogeneous nucleation agent, which increase the ratio of crystalline region and amorphous region interface area in polymer, there are plenty of cavity traps within interface area. Secondly, the doping nanoparticles and polymeric matrix form an interface area. The small-size and high surface energy of nanoparticles cause the stronger interaction between nanoparticles and matrix resin. The ordering, motility and free volume of macromolecule chain segment arrangement are different from which of other high polymer chain in this area, which makes trapping action and scattering process to carrier. Besides, nanoparticles doping could improve the polymer traps level density evidently. With the nanoparticles doping, the deep traps are introduced and the carrier is hard to make detrapping after trapping. The interface localized state is formed within the samples. On the one hand the charge trapping diminish the carrier drift velocity, on the other hand the Coulomb field, which is produced by like charges, offset partial external electric field. It can reduce the local electric field intensity within samples. The charge mobility decreased with the quantity of electric charge injected by electrode. By the effects of these factors, the carrier quantity and mobility of nanocomposite fell drastically, which improve the composite conductivity. There are no traps in inorganic ZnO particles, whereas the traps existed in interfacial structures produced by the combination of ZnO with LDPE.

6.       The conductivity of ZnO/LDPE material is limited to the interaction between powders. With the ZnO particles size decreased, the interface quantity increased and conductivity decreased. Because the conductivity of ZnO is larger than which of interfacial structure, the electric field applied to interface mainly. The quantity of unit thickness interface decreased with particle size increased. Besides, the electric field intensity endured by each interface is stronger. Therefore the conductivity of large particle ZnO material is increasingly dependent on electric field intensity. The electric theoretical model of ZnO/LDPE composites is “filler-interface-polymer”. Among them the composites electrical conductivity is mainly affected by the interface and polymeric matrix. With the increase of electric field intensity, the composites conductivity property is controlled by interface, which is converted from polymeric matrix. The thermal stimulus jump electrical conductivity is reflected in composites conductivity of polymeric matrix control mode and interface control mode. Therefore the carrier of composite conductivity is the recombination of electron and ion. From the experimental research of DC electrical conductivity characteristics, it can be seen that: The inorganic filler particles size show a certain percolation characteristics. The inorganic filler has an effect on the percolation characteristics of composites conductivity. With the same mass fraction nano-ZnO doping, the ZnO particles size decreases, the particles quantity would increase and the conductivity nonlinearity is higher in composite. Therefore the effect of different particles size nano-ZnO doping on composites electrical conductivity characteristics is different.

7.       The English in this article has been revised.

8.       Particles polarity: the nano-ZnO particles is inorganic particles, of which surface include organic group. After the surface finish, the nanoparticles are combined with polymeric matrix betterElectric field intensity: Under uniform electric field, the ratio of electric voltage and dielectric thickness is electric field intensityThe relative volume of interface: It is the ratio of interfacial structure in composite. In unit volume of polymeric composites, the interfacial structure volume grew larger with nanoparticles size decreases.

9.       The melting blend was used to prepare the nano-ZnO/LDPE composites in this article. Above the viscous flow temperature of ZnO and LDPE, the rotational speed rheometer was used to prepare the uniform polymeric co-melt which was cooled then. In this article, the rotational speed rheometer was used to make the melt blending of nano-ZnO with LDPE. The temperature of rotational speed rheometer was set to 140 and the duration is 20min.

Reviewer 2 Report

Dear authors, 

Please find attached a document with the main comments and suggestions to your article. 

I hope you find it useful.

Author Response

Author Response:

First of all, I’d like to say thanks to the expert’s comments on the manuscript. I feel honored to gain the expert’s encouragement. I agree with the expert on the comments and revision suggestion. I have modified the manuscript and improved the language of the manuscript, the detailed modifications are shown as follows:

Detailed Modifications:

1.       The title has been revised, which is “Effect of Particles size on Dielectric Properties of Nano-ZnO/LDPE Composites”.

2.       The amount of ZnO doping is 3wt% in this article, which is marked in abstract, samples preparation and comment section of figure 1.

3.       “The scanning electron microscope (SEM) and differential scanning calorimeter (DSC) could observe the ZnO particle size and ZnO/LDPE nanocomposite crystallization” is corrected to “Scanning electron microscopy (SEM) was used to characterize ZnO nanoparticles whereas differential scanning calorimetry (DSC) was used to do thermal characterization of the samples”.

4.       The crystalline characteristic has been deleted in key words.

5.       The introduction has been revised, which is divided into two parts. The research background and significance, the main research contents, methods and goals of this article are discussed respectively.

6.       The “high apparent activation energy” refers to nanoparticles surface effect. When the particles size reduce to nanometer level, the ratio of material surface atoms quantity and integrity crystalline grain atoms quantity increased. Owing to the increase of nanoparticles surface atoms quantity, the atoms coordination is insufficient. A great deal of dangling bonds exists, which raise the surface chemistry activity.

7.       “As the nanoparticles size decreasing, the specific area increases and the apparent activation energy rise,…” is corrected to “As the nanoparticle size decreases, the specific area increases and the apparent activation energy rises,…”.

8.       The amount of ZnO doping is 3wt% in this article, which is marked in abstract, samples preparation and comment section of figure 1.

9.       The spelling error has been revised in this article.

10.   The comment section of figure 2 has been revised, in which b) 100 nm is corrected to c)100nm.

11.   2.3 heading is corrected to crystallization analysis of ZnO/LDPE composites.

12.   DSC experiment has been implemented based on the expert opinion. The heating and cooling DSC curve of composites is added into this article. The symbols and units are corrected to Heat flowa.u.. Meanwhile, the endothermic arrow is marked out. The comment section of figure 2 has been corrected to” DSC traces of the composite materials”.

13.   The starting temperature is 25 in DSC experiment, which is marked in 2.3.

14.   Line 101 “from which it can be obtained”.

15.   The fast response DSC is used in isothermal crystallization process. The melting state sample was chilled to a certain temperature (crystallizing point) under the melting point, which is measured in constant temperature.

16.   In table 1, the “particles size of filler” is replaced by “sample”.

17.   The nano-ZnO particles size is 9nm, 30nm, 100nm and 200nm respectively in this article.

18.   The Weibull distribution curve of composites breakdown field strength is shown in figure 4, figure 5 and figure 6. The materials breakdown probability under a special electric field intensity E and the failure probability within a certain time t could be reflected. The result of research shows that the Weibull distribution statistical is ideal for the analysis of insulation material breakdown field strength. The parameters in Weibull distribution represent the form parameter β of data spread and Weibull breakdown field strength E. The breakdown field strength value of 20 samples is passed by the line of nano-ZnO/LDPE composites with different particles size. The data is analyzed by minitab statistical software. The annotation in these figure are described in English, among them N represents the test times of each sample.

19.   The samples thickness is marked in figure 8, which is 65±10μm.

20.   The reference about the space charge limited current theory is added into this article.

Reviewer 3 Report

The manuscript is well written and result clearly presented. Minor changes of English language and grammar are required.

Manuscript title: " The research of Crystallization and Dielectric Propert in different particles..."; did you mean Dielectric Properties? please correct

Authors should describe the aims and the potential applications. 

Author Response

Author Response:

First of all, I’d like to say thanks to the expert’s comments on the manuscript. I feel honored to gain the expert’s encouragement. I agree with the expert on the comments and revision suggestion. I have modified the manuscript and improved the language of the manuscript, the detailed modifications are shown as follows:

Detailed Modifications:

1.       The title is “Effect of Particles size on Dielectric Properties of Nano-ZnO/LDPE Composites”.

2.       The research contents in this article is mainly applied in cable major insulation material, Low density polyethylene (LDPE) is one kind of nonpolar macromolecule polymeric material which has some advantages such as high insulation resistance, low permittivity, low loss and low impact of temperature. Therefore, LDPE is widely used as base material of electric power cable insulation system. Nano-ZnO is one kind of new inorganic product. It is an elaborate compound semiconductor material with high performance and large energy gap, and it has  nonlinear characteristics. Compared with common ZnO, It has many advantages such as surface effect, volumetric effect, quantum effect and dielectric confinement effect. Thanks to the excellent properties of nano-ZnO, it is used widely. According to the blending technology, the inorganic semiconductor material with nonlinear characteristic could be added into the polymeric matrix. The nolinear composite insulating material of which conductivity and dielectric constant changed with electric field intensity could be prepared. This nonlinear insulation material has the function of homogenizing the electric field in complex insulation structure, which could improve the dielectric strength of insulation system. Therefore it has a broad space such as being used in insulation system of extreme high voltage DC bushing and ultrahigh voltage direct current cableTherefore , the dielectric property of ZnO/LDPE composites is researched in this article.

Round 2

Reviewer 1 Report

I agree with Yours corrections and revisions.

To my mind English still can be edited a little. But I am not English speaking.

Author Response

First of all, I’d like to say thanks to the expert’s comments on the manuscript. I feel honored to gain the expert’s encouragement. I agree with the expert on the comments and revision suggestion. I have modified the manuscript and improved the language. Besides, the detailed modifications are marked in red.

Reviewer 2 Report

Dear authors, 

Please find attached a document with some comments and suggestions for this version of the manuscript. I hope that the revised version will be improved with them. 

Author Response

First of all, I’d like to say thanks to the expert’s comments on the manuscript. I feel honored to gain the expert’s encouragement. I agree with the expert on the comments and revision suggestion. I have modified the manuscript and improved the language. The detailed modifications are shown as follows:

1、  The English in this article has been revised. For example, the sentence “nano-ZnO/LDPE composites has” was revised to “nano-ZnO/LDPE composites had”.

2、  The introduction was modified and improved further.

3、  Regarding to figure 2, we have picked a set of relatively clear SEM diagrams under current experimental condition.

4、  “Recording the enthalpy change could obtain the crystal temperature Tc and the width of exothermic crystalline peak DTc could be obtained” was revised to “The enthalpy change was recorder, from which the crystal temperature Tc and the width of exothermic crystalline peak DTc could be obtained”.

5、  The related reference has been added to illustrate the sentence of “The higher the melting temperature, the bigger the spherulite size was”

6、  The related reference has been added to explain the H0 of LDPE.

7、  After different size nano-ZnO particles doping, the melting temperature of nano-ZnO/LDPE composites decreased. It also explained that nanoparticles doping would diminish the crystal size within samples.

8、  Regarding to figure 4, the description of β, E and N has been added into the article.

9、  The text in lines 206-225 has been rewritten.

All the modifications in this manuscript have been marked in red.
